# Dietary Supplementation of Capsaicin Enhances Productive and Reproductive Efficiency of Chinese Crossbred Buffaloes in Low Breeding Season

**DOI:** 10.3390/ani13010118

**Published:** 2022-12-28

**Authors:** Adili Abulaiti, Zahid Naseer, Zulfiqar Ahmed, Wenju Liu, Xunsheng Pang, Muhammad Farooq Iqbal, Shujuan Wang

**Affiliations:** 1College of Animal Science, Anhui Science and Technology University, Fengyang 233100, China; 2Anhui Province Key Laboratory of Animal Nutritional Regulation and Health, Fengyang 233100, China; 3Faculty of Veterinary and Animal Sciences, Pir Mehr Ali Shah Arid Agriculture University, Rawalpindi 46000, Pakistan; 4Key Laboratory of Swine Genetics and Breeding of Ministry of Agriculture and Rural Affairs, and Key Lab of Agricultural Animal Genetics, Breeding and Reproduction of Ministry of Education, Huazhong Agricultural University, Wuhan 430070, China; 5College of Life and Health Science, Anhui Science and Technology University, Fengyang 233100, China

**Keywords:** capsaicin, crossbred buffaloes, synchronization, production, fertility, season

## Abstract

**Simple Summary:**

Heat stress affects the productivity of exposed animals by lowering fertility-related aspects. This phenomenon is more pronounced in buffaloes, particularly during the low breeding season, as it is coupled with thermal assaults. Under such conditions, altered metabolic activity is markedly noticed, affecting the production and reproduction of buffaloes. The obtained results revealed that the use of capsaicin is beneficial to improve milk yield and composition without altering the other metabolic process. Moreover, improved reproductive performance is achieved in buffaloes when capsaicin is supplemented prior to synchronization.

**Abstract:**

The present study investigated the role of dietary capsaicin (CPS) supplementation on milk yield (liters/head) and milk composition (total solids, lactose, albumin, protein, fat, milk urea nitrogen (MUN), somatic cell count (SCC) and somatic cell score (SCS), serum metabolites (lipoprotein esterase (LPL) and aspartate aminotransferase (AST)), and reproductive physiology (follicular development, estrus response, ovulation and pregnancy) following synchronization during the low breeding season. One hundred (*n* = 100) crossbred buffaloes were randomly assigned to four dietary groups consisting of CPS supplementation dosages (0, 2, 4 or 6 mg/kg of total mixed ration; TMR) as CPS-0 (*n* = 26), CPS-2 (*n* = 22), CPS-4 (*n* = 25) and CPS-6 (*n* = 27), respectively, in a 30-day feed trial. The results revealed that the CPS-4 group of buffaloes had a better estrus rate (72%) along with improved (*p* < 0.05) ovulatory follicle diameter (13.8 mm), ovulation rate (68%) and pregnancy rate (48%) compared to other treatment groups. Milk yield improved (*p* < 0.05) in CPS-4 supplemented buffaloes after day 20 of the trial, comparatively. There was a significant effect (*p* < 0.05) of milk sampling day (day 30) on total milk solids, lactose, milk protein and MUN levels, whereas lactose, MUN, SCC and SCS were influenced by supplementation dosage (CPS-4). Glucose levels were affected in buffaloes by sampling time (artificial insemination (AI) and 50-day post-AI) and CPS-dose (CPS-4 and CPS-6), respectively. LPL level changed in CPS-2 and CPS-4 groups at AI time and 50 days after AI. In addition, the AST level was different in CPS-4 at AI time and 50 days after AI. Therefore, our data suggest that a medium dose (~4 mg/kg of TMR) of CPS provided a better response in the form of milk yield, milk composition, serum metabolites and reproductive performance in crossbred buffaloes during the low breeding season.

## 1. Introduction

Buffalo production is advantageous due to its ability to survive by utilizing the poor quality forage and adapting to harsh environmental conditions [1]. However, regarding reproductive issues in buffaloes, late sexual maturity, high incidence of anestrous in summer, longer post-partum interval, subtle estrus expression and poor conception rate post-AI significantly reduced the herd profitability [2]. In the summer of the tropics and subtropics, thermal assault is also a lethal factor in buffalo reproductive inadequacy and breeding pattern [3]. Summer heat stress conditions lead to buffalo inadequate productive and procreative performance because of altered animal feeding behavior and metabolic activities. In addition, increased peripheral and reduced internal blood flow under heat stress periods affect the physiology of the digestive, mammary gland and reproductive systems, lowering the productive and reproductive efficiency of exposed animals [4]. To cope with adverse productive and reproductive issues, increased feed intake and enhanced metabolic rate are necessary for buffaloes under summer stressed conditions.

To improve the reproductive efficiency of buffaloes, particularly in the low breeding season or heat stress conditions, various estrus and ovulation synchronization regimens are in practice [5,6,7,8]. The Ovsynch protocol is a popular regimen in cow dairy herds for timed AI; however, its usage is modest in buffalo herds, particularly in summer when limited cyclic buffaloes are available. Low fertility following the Ovsynch protocol is usually linked to the reduced ovarian activity of buffaloes under summer-stressed conditions. Moreover, low metabolic activity due to decreased feed intake could be an accompanying element for low productive and productive efficiency in buffaloes [9,10]. Therefore, direct or indirect dietary energy supplementation can potentially enhance the metabolic and endocrine milieu for ovarian functioning [11]. For this reason, CPS has been identified as a potential nutraceutical to promote blood circulation towards the digestive system from surface circulation. This phenomenon leads to improved feed intake and metabolic activities that could ensure the availability of vital nutrients for milk production, endocrine functioning and embryo survival [9,10,11,12].

Capsaicin is a derivative of capsicum, comprised of a phenolic structure. The beneficial effects of CPS effects as a rumen tonic [13], digestive enzyme stimulant [14] and glucose homeostasis [15] have been observed earlier in dairy animals. Previously, CPS has been tested in heat-stressed cows, where it improved metabolic, productive [10,16] and reproductive performance [11]. Like cows, CPS as a feed supplement would be an ameliorative strategy to enhance the metabolic activity for milk production and the effectiveness of fixed-time AI protocol for buffaloes. As aforementioned, CPS, loaded with different biological features, might be effective in enhancing reproductive and productive performance, particularly in stressful conditions of the summer season. The present study postulated that using CPS along ovulation synchronization protocol seems a potent program to mitigate heat stress linked to productive deficiencies in buffaloes. This study is intended to elucidate the role of CPS as photogenic supplements on the productive and reproductive efficiency of crossbred buffaloes during thermal stress conditions.

## 2. Materials and Methods

### 2.1. Climatic Conditions of Experimental Site

This research was performed at a private buffalo farm (Hubei Jinniu Co. Ltd., Jingzhou, China) in Hubei province, China (latitude 30°32′ N, longitude 111°51′ E). In the study area, buffalo breeding is differentiated into peak (from September to November end) and low (from June to the end of August) breeding seasons. This experiment was carried out during months (July to August) of the low breeding season when peak ambient temperatures (35 to 39 °C) and relative humidity (80% to 90%) are observed. Daily ambient temperatures (T °C) and relative humidity (RH) of the experimental site were recorded (08:00 h, 14:00 h and 18:00 h) during the study period. Temperature-humidity index (THI) was calculated using the following formula for assessing the heat stress intensity [16]:THI = (0.8 × T °C) + [(RH/100) × (T °C − 14.4)] + 46.4 

During the study period, average T (33.9 ± 2.3 °C), RH (83.4± 7.3%) and THI (91.4 ± 2.4) were observed. The observed THI values (91.4 ± 2.4) corresponded to severe heat stress for buffaloes (Figure 1).

### 2.2. Animal Husbandry Practices

For this purpose, one hundred (*n* = 100) multiparous freshly calved (2.9 ± 0.5 days in milk; DIM) crossbred buffaloes (Mediterranean × Murrah × Nili Ravi × Jianghan) with average body weight 591.6 ± 75.8 kg and good body condition score (3.27 ± 0.68), were selected. The buffalo farm comprised an open shed with a cemented rooftop along fenced side walls by galvanized wire mesh. In addition, the facility of exhaust fans and sprinklers was provided for maintaining the humidity level and air quality during summer days. Buffaloes were kept under a tail-to-tail stall feeding system with an area of ~3.7 m^2^ along a manger area (0.6 × 0.9 m). Buffaloes were supplied with fresh and clean water ad libitum. Machine milking was practiced for milking the buffaloes twice a day. All the animals were in good health with physical conditions nearly of the same body size in each group.

### 2.3. Treatment Feed Formulation and CPS Supplementation

The buffaloes were usually fed with a total mixed ration (Table 1), which was analyzed for total digestible nutrients, crude protein, crude fiber, and ether extract through proximate analysis prior to feeding according to the AOAC procedure [17]. The selected buffaloes (*n* = 100) were divided into four experimental CPS (Capsaicin; Guangzhou Pumai Biotechnology, China) dosage (0, 2, 4 and 6 mg/kg of TMR) groups. CPS was mixed properly into the TMR at a specified ratio (0, 2, 4 or 6 mg/kg) and TMR was offered to the respective group (CPS-0 (*n* = 26), CPS-2 (*n* = 22), CPS-4 (*n* = 25) or CPS-6 (*n* = 27)) daily for 30 days starting from 20th day of calving. In the first half of feeding time, a ration loaded with CPS dose (0, 2, 4 or 6 mg/kg) was offered and later remaining portion ration was provided to each buffalo. Prior to offering the second half of the ration, it was ensured that every buffalo had no leftover of CPS supplemented ration in the manger.

### 2.4. Synchronization Protocol and Estimation of Reproductive Parameters

The buffaloes were monitored for regular estrous cycles by tracking follicular development and ovulation on a weekly ultrasonography basis. Following the CPS supplementation (50th day of calving or Day 0 of protocol), the buffaloes were treated with gonadotropin-releasing hormone (GnRH, 400 µg, intramuscular (IM); Ningbo Sansheng Pharmaceutical (NSP), China) at day 0 and an injection of prostaglandin F_2α_ (PGF_2α_, 0.5 mg, IM; NSP) at day 7 of the protocol. The second injection of GnRH (400µg) and mifepristone (0.4 mg/kg, IM; Hubei Yun Cheng Sai Technology, China) was administered on day 9 of the protocol. Timed-AI using cryopreserved semen was performed after 16 h of the second injection of GnRH in synchronized buffaloes. At day 5 of post-AI, human chorionic gonadotropin (hCG, 2000 IU, IM; NSP) was injected intramuscularly into each inseminated buffalo (Figure 2; [18]).

Buffaloes were scanned for monitoring the follicular dynamics twice daily (starting from Day 6 to Day 12 of protocol) through a veterinary ultrasound scanner (WED-9618-v; Shenzhen Well D Medical Electronics, Guangdong, China) with a rectal probe. The ovulation process was recorded based on the disappearance of the ovulatory follicle observed through subsequent ultrasonography scans with an interval of 12 hrs [19]. Estrus response was based on detecting estrus signs (vaginal mucous discharge, bellowing, milling, swollen vagina, and head butting) through visual observations (06:00 h and 18:00 h). Buffaloes were diagnosed with the pregnancy on the 50th day of the protocol using a transrectal ultrasound scanner.

### 2.5. Milk Production and Composition Traits

Following each milking (06:00 h and 16:00 h), daily milk yield was recorded and measured in liters using calibrated jars at intervals of 5 days (start of CPS supplementation to Day 45 of the trial). For the analysis of milk composition, individual milk sample (a composite sample from morning and evening milking) was collected on the same day of milk yield recording (days 20 and 30 of the trial). A milk composition analyzer (type 78110; Foss, Denmark) was used to determine the milk composition parameters (total solids, lactose, fat, protein, SCC, SCS and MUN).

### 2.6. Estimation of Metabolic Activity

Serum glucose, LPL and AST levels were estimated following blood sampling from each buffalo group at the start of CPS supplementation, time of AI, and 50 days after AI, respectively. Blood glucose level was measured through a bovine blood glucose test kit (Ml076792). LPL and AST levels were estimated by bovine lipoprotein esterase test kit (Ml607622; sensitivity 3 µg/mL, intra-assay variation <15%, interassay variation <15%) and aspartate aminotransferase test kit (Ml063776; sensitivity 0.5 ng/mL, intra-assay variation <8%, interassay variation <10%).

### 2.7. Statistical Analyses

Data (Mean ± standard deviation) were analyzed using statistical software (SPSS version 17.0.1 Chicago, IL, USA). The normality of whole data distributions was evaluated. One-way analysis of variance (ANOVA) was used to compare the reproductive variables (follicular dynamics, duration to estrus or ovulation) among groups. Data of pregnancy, estrus and ovulation across the groups were compared through the *chi*-square test using the Prism-6 software package (GraphPad Software, San Diego, CA, USA). Repeated measures ANOVA was used to compare the main effects (dosages and sampling day) and their interactions for metabolic parameters (glucose, ALT and AST) milk production variables (milk yield, total solids, lactose, albumin, protein, fat, urea nitrogen, somatic cell count and somatic cell score) of buffaloes. The value of *p* < 0.05 was considered significant.

## 3. Results

### 3.1. Effect of Capsaicin on Reproductive Indices in Synchronized Buffaloes

The results in Table 2 indicate that the CPS-4 group of crossbred buffaloes showed significantly (*p* < 0.05) better estrus response than other treatment groups. In comparison, there were no differences (*p* > 0.05) in the interval to estrus after the second GnRH and estrus duration in different dosages of CPS. A higher number (*p* < 0.05) of ovulations were observed in CPS-4 supplemented buffaloes compared to control or CPS-2 and CPS-6 supplemented synchronized buffaloes. The diameter of the ovulatory follicle was also greater (*p* < 0.05) in the CPS-4 treatment group than in the CPS-6 group. Interval to ovulation after the second GnRH was not different amongst the treatment groups. The pregnancy rate was improved significantly (*p* < 0.05) in the CPS-4 buffalo group than the CPS-0 or CPS-2 groups. The CPS or synchronization protocol did not influence the growth rate and size of developing follicles on day 6 of the protocol in buffaloes (Table 1).

### 3.2. Milk Yield of Crossbred Buffaloes Supplemented by Different Doses of CPS

The effect of CPS supplementation on milk production is presented in Figure 3. Milk yield did not improve in any group of buffaloes during the initial two weeks of CPS supplementation. Milk production gradually increased (*p* < 0.05) from day 20 onwards within each group, whereas maximum improvement (*p* < 0.05) in milk production was recorded in CPS-4 supplemented group as compared to CPS-0 and CPS-2 supplemented buffaloes.

### 3.3. Effects of CPS Supplementation on Composition Milk Produced by Buffaloes

The results in Table 3 showed the effects of CPS dose level and sampling time on milk composition traits. No interaction was found for milk composition traits; therefore, the main effects were presented. A significant change was recorded in total solids in milk produced by the buffaloes supplemented through 4 and 6 mg CPS dose levels at days 20 and 30 of observations. A similar protein level was observed in milk produced by CPS-supplemented and non-supplemented buffaloes, but its level changed between days 20 and 30 of observation. Milk fat levels were neither influenced by day supplementation nor CPS dose. There were significant variations in lactose levels (*p* < 0.05) for CPS dose and day of observations. A higher milk protein level (*p* < 0.05) was noted at day 30 compared to day 20 of supplementation in buffaloes but was not affected by CPS dosage. The MUN level varied (*p* < 0.05) in CPS supplementation dose in buffaloes and day of observation. The SCC and SCS levels were significantly (*p* < 0.05) different across the CPS-supplemented groups; however, no alteration was observed by the day of observation.

### 3.4. Effects of CPS Supplementation on Serum Metabolites of Crossbred Buffaloes

Serum metabolites presented in Table 4 indicate the main effects of CPS dosage and the day of sampling. No interaction was observed between the CPS treatment group and the sampling day. The glucose level was similar amongst the groups at the start of CPS-supplementation; however, glucose level improved significantly (*p* < 0.05) in CPS-4 and CPS-6 supplemented groups at the time of AI and 50 days post-AI. No fluctuation occurred in LPL and AST levels of CPS-supplemented buffaloes, while CPS-2 and CPS-4 buffaloes had higher (*p* < 0.05) LPL levels at the time of AI and 50 days post-AI compared to the start of the trial. In the case of AST, only the CPS-4 group had variations (*p* < 0.05) at different observation time points.

## 4. Discussion

Capsaicinoid compounds cause the spicy flavor in chili pepper fruit. Dihydrocapsaicin and CPS are primary capsaicinoids that share ~90% chili pepper fruit. The CPS (trans-8-methyl-*N*-vanillyl-6-nonenamide) is an odorless and colorless crystal with a lipophilic alkaloid nature compound. As an active compound, CPS is used in the pharmaceutical and food industry for chemical and enzyme production. It has vast applications in biological and physiological conditions, such as antioxidant, anti-carcinogenic, anti-inflammatory and energy promoters [20]. CPS has also been tested in animal production to improve productivity by promoting different physiological mechanisms such as nutrient mobility, rumen tonicity, increased blood flow to internal organs and fertility enhancers [9,10,11]. The present study connects with a previous report [11] that describes the potential effect of CPS supplementation on milk production and fertility events of buffaloes under thermal assaults.

The present study demonstrates that CPS supplementation improved the effectiveness of the modified Ovsynch protocol by improving the pregnancy rates (48% in CPS-4) in buffaloes in the low breeding seasons. Earlier, poor pregnancy rates were reported following Ovsynch protocol in buffaloes during the low breeding season [21]. The obtained pregnancy rates in synchronized buffaloes might be linked to CPS supplementation in the experimental animals. One aspect of CPS supplementation and enhanced fertility is explained that it binds to receptors in the plasma membrane and activate ligand-gated, nonselective cation channels for the excitation of sensory neurons [22,23] and CPS-excited sensory neurons, which in turn improve granulosa cells proliferation [24], folliculogenesis and fertility in different species [11,25,26]. Increased metabolism and availability of nutrients, particularly glucose, at breeding and during the initial stages of pregnancy by CPS supplementation might be an involved mechanism for the enhancement of fertility in buffaloes during the low breeding season.

The effectiveness of the Ovsynch protocol is based on the selection of cyclic animals with follicles of appropriate size ( >8 or 9 mm) at the first injection of GnRH in the Ovsynch protocol [27]. The mentioned criteria were followed in the present study by pre-monitoring of ultrasonographic tracking; hence, an ideal group of buffaloes was submitted for modified Ovsynch protocol application. In the present study, fertility outcomes were comparable to earlier reports when buffaloes were supplemented with flaxseed prior to Ovsynch initiation [28]. Obtained pregnancy rates were higher compared to previous reports where conventional Ovsynch protocol was practiced during the low breeding season [21,29,30]. The inclusion of mifepristone and hCG injection seems a beneficial incorporation in Ovsynch for improvement of pregnancy rate in buffaloes during the low breeding season. Similar observations were made in previous reports on buffaloes [31,32]. It is expected that treatment with hCG on day 5 after AI might increase the progesterone level by promoting secretion from the primary corpus luteum by inducing the luteinization or ovulation of existing follicles to an accessory corpus luteum. In the future, CPS supplementation prior to different synchronization protocols in fertility-compromised buffaloes needs to be investigated across peak and low breeding seasons.

The current study revealed that using CPS partially alleviated the negative effects of high ambient temperature and humidity on the productive performance of buffaloes. Milk yield was highly promising when a medium dose (~4 mg/kg of TMR) was offered to buffaloes in summer conditions. Improvement in milk yield was recorded after two weeks of supplementation, indicating that a certain period of CPS supplementation is required for animals to alleviate summer stress conditions. Enhanced milk production in the present study might be associated with altered rumen fermentation patterns [33] either through altered rumen microbial biota or improvements in rumen/gut health status by CPS supplementation [34]. Previous studies have shown the diversity in milk production responses following CPS supplementation under normal [33,35,36] and stress conditions [9,10,11] in cows. Determining dry matter intake and digestibility could reveal the direct effect of CPS on animal metabolic efficiency, as An et al. [10] reported in heat-stressed cows. Occurrence of variability in different reports is more likely to be linked with duration, dose or form of CPS supplementation; stage of production; and route of administration to submitted animals. However, the effect of CPS on the change in ruminal microbiota at different intervals and nutritional profile linked to milk yield under normal and stressed conditions needs to be elucidated in ruminants.

In the present study, CPS supplementation has not been associated with an effect on milk components of buffaloes under summer-stressed conditions except for lowered MUN and SCC after one month of CPS supplementation. There are different available reports concerning milk composition in cows that indicate the variation in milk fat [10,11,33,35] and lactose contents [10] or no change in milk contents [12,15,36] after CPS supplementation. Lowered milk urea nitrogen in CPS groups indicated a changed ruminal microflora in response to CPS supplementation that promoted the utilization of ammonia; in turn, limited urea nitrogen was available for entering into the bloodstream and milk [37], whereas increased MUN level in other groups could be associated to increased catabolic activities of body tissues under heat stress [38]. The decreasing trend in SCC may be related to low pH levels because of low MUN levels following CPS supplementation in cows. As the pH is inversely linked to SCC in milk [39], providing CPS can be a good source to minimize mastitis incidence in buffalo herds.

The current findings of serum glucose and liver enzymes (LPL and AST) depict the beneficial impact of CPS supplementation in buffaloes under stress. The increasing glucose trend at AI and post-AI reflected the dietary glucose absorption, hepatic output, and tissue glucose clearance during the trial [38]. It has also been observed that a group of buffaloes receiving a medium dose of CPS showed better fertility results which might be linked to increased availability of glucose at breeding and post-breeding times [10]. The low glucose in counterparts might be due to the low or no effect of CPS to overcome the heat stress in buffaloes. As glucose is a precursor for lactose synthesis, we did not notice any significant variation in milk lactose levels among the groups. Numerically higher lactose contents are linked to the modulation of CPS supplementation in buffaloes under stressed conditions [34]. The liver is the organ that contains the most abundant enzymes in an animal body and accounts for about 2/3 of the total protein content of the liver. The changes in serum enzyme activity reflect the pathological state of the liver [40], but, presently, no change was observed in ALT and AST levels and remained constant at each observation point. Liver activity shows that CPS supplementation is safer and does not interact with the normal functioning of the liver. There are numerically higher values of ALT and AST in CPS-supplemented buffaloes; however, these values do not correspond to the pathological conditions as observed earlier [11]. This study affirms that CPS supplementation provides better fertility and production outcomes without affecting the liver integrity in buffaloes under stress conditions.

## 5. Conclusions

In conclusion, CPS supplementation can provide better responses in the form of milk production and maintain the metabolic function of crossbred buffaloes during the heat exposure period. Moreover, a medium dose (~4 mg/kg of TMR) of CPS can enhance reproductive performance in crossbred buffaloes during the low breeding season.

## Figures and Tables

**Figure 1 animals-13-00118-f001:**
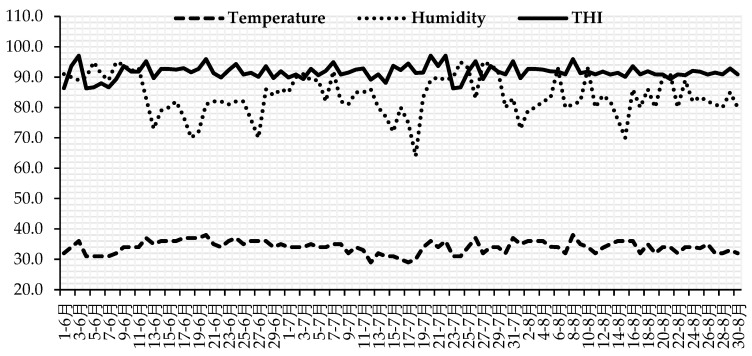
The observed daily variations in temperature (°C (------ hyphen)), relative humidity (% (•••••••••••dotted line)) and THI (______solid line) during the experiment.

**Figure 2 animals-13-00118-f002:**
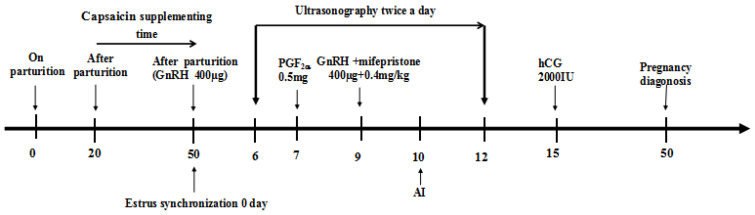
The experimental diagram represents the schedule for CPS provision and modified Ovsynch synchronization protocol for crossbred buffaloes during the low breeding season. Buffaloes were supplemented with CPS (0, 2, 4 or 6 mg per kg of TMR) for 30 days (starting from day 20 of calving) and synchronized through modified Ovsynch synchronization protocol (Day 0 = first GnRH injection, Day 7 = PGF2α injection, Day 9 = Second GnRH and mifepristone injections, Day 10 = TAI, Day15 = hCG injection and Day 50 = Pregnancy diagnoses). Follicular dynamics were monitored from 6 to 12 days of the protocol through ultrasonography in synchronized buffaloes (adapted from Abulaiti et al. [18]).

**Figure 3 animals-13-00118-f003:**
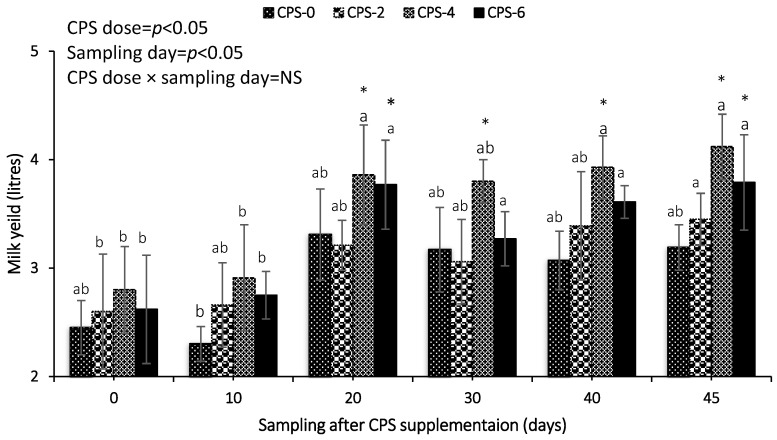
Comparison of milk yield (liters; mean ± S.D.) of buffaloes supplemented with different CPS doses (0, 2, 4 and 6 mg/kg TMR) for a duration of 45 days. * Difference (*p* < 0.05) between CPS groups on a sampling day. ^a,b^ Within each CPS group over sampling duration, means without a common superscript differed (*p* < 0.05). NS means non-significant.

**Table 1 animals-13-00118-t001:** Ingredients and proximate composition of TMR.

Ingredients Composition	
Ingredients	Feed Basis; %
Wheat straw	19
Maize silage	20
Peanut hay	4
Distiller grains	12
Camellia oil powder	5
Rape seed cake	8
Corn grain	20
Corn gluten 60%	8
Urea	0.5
Monosodium phosphate	0.5
Salt	0.5
Limestone	2
Premix	0.1
**Proximate composition of TMR**	
Total digestible nutrient (%)	69
Crude protein (%)	17.5
Crude fiber (%)	17.72
Ether extract (%)	5.62

**Table 2 animals-13-00118-t002:** Reproductive variables in buffaloes treated with modified Ovsynch regimen after supplementation of CPS (0, 2, 4 or 6 mg per kg of TMR) during low breeding season.

Variables	CPS-0 (*n* = 26)	CPS-2 (*n* = 22)	CPS-4 (*n* = 25)	CPS-6 (*n* = 27)
Diameter of first detecting follicle (mm)	8.2 ± 1.7	8.9 ± 1.9	8.7 ± 1.9	8.6 ± 1.6
Estrus response (%)	50 (13/26) ^b^	63.6 (14/22) ^ab^	80.0 (20/25) ^a^	62.9 (17/27) ^ab^
Interval to estrus after 2nd GnRH (h)	9.7 ± 3.2	8.9 ± 2.1	8.5 ± 2.7	8.7 ± 2.7
Estrus duration (h)	16.4 ± 1.5	16.0 ± 1.3	15.7 ± 1.7	16.8 ± 1.1
Ovulation rate (%)	46.2 (12/26) ^b^	59.1 (13/22) ^ab^	76.0 (19/25) ^a^	55.6 (15/27) ^ab^
Interval to ovulation after 2nd GnRH (h)	25.3 ± 1.9	25.5 ± 2.0	24.5 ± 1.4	25.8 ± 1.6
Ovulatory follicle diameter (mm)	12.7 ± 1.6 ^ab^	12.0 ± 1.9 ^b^	13.8 ± 1.6 ^a^	11.9 ± 1.3 ^b^
Follicle growth rate (mm/day)	1.5608	1.4677	1.5408	1.4517
Pregnancy rate (%)	19.2 (5/26)^b^	36.4 (8/22) ^ab^	48.0 (12/25) ^a^	40.7 (11/27) ^ab^

Note: Lowercase letters (^a,b^) in the same row indicate significant differences among the treatment groups (*p* < 0.05).

**Table 3 animals-13-00118-t003:** Milk composition (Total solids, lactose, albumin, protein, fat, urea nitrogen, somatic cell count and somatic cell score) of buffaloes after 20 and 30th day of CPS (0, 2, 4 or 6 mg/kg of TMR) supplementation.

Variables	Supplementation Day	CPS-0	CPS-2	CPS-4	CPS-6	CPS Dose	Supplementation Day
Total solids (%)	20	17.03 ± 1.5	18.2 ± 1.7	18.45 ± 1.4	18.96 ± 1.8	0.259	<0.0001
	30	17.67 ± 1.8	17.41 ± 2.2	19.09 ± 1.9	19.81 ± 1.5		
Lactose (%)	20	4.65 ± 0.4	5.31 ± 0.3	5.29 ± 0.3	5.50 ± 0.5	<0.0001	<0.0001
	30	4.70 ± 0.2	6.32 ± 0.7	6.80 ± 0.6	6.58 ± 0.7		
Milk protein (%)	20	4.02 ± 0.8	4.13 ± 1.3	4.27 ± 1.3	4.14 ± 0.8	0.170	<0.0001
	30	4.14 ± 1.3	5.20 ± 0.8	5.29 ± 0.6	5.12 ± 0.9		
Milk fat (%)	20	6.68 ± 1.3	6.41 ± 2.2	6.58 ± 2.6	6.88 ± 1.5	0.302	0.230
	30	6.46 ± 1.1	7.35 ± 1.7	7.63 ± 1.5	7.75 ± 1.2		
Milk urea nitrogen (mg/dL)	20	11.10 ± 1.1	12.67 ± 1.3	11.42 ± 1.2	12.36 ± 1.4	<0.0001	<0.0001
	30	12.93 ± 1.2	10.30 ± 1.4	8.35 ± 1.0	11.12 ± 0.7		
Somatic cell count (10^3^/mL)	20	65.65 ± 9.6	76.46 ± 10.1	55.76 ± 13.3	55.52 ± 10.6	<0.0001	0.190
	30	95.9 ± 7.1	75.25 ± 10.7	54.84 ± 10.3	56.44 ± 8.5		
Somatic cell score (*n*)	20	5.82 ± 1.6	4.05 ± 1.7	3.9 ± 2.3	4.05 ± 1.3	<0.0001	0.920
	30	5.83 ± 2.0	4.05 ± 1.5	3.72 ± 1.2	4.11 ± 1.5		

**Table 4 animals-13-00118-t004:** Serum glucose, LPL and AST levels in buffaloes after different supplementary doses of CPS (0, 2, 4, and 6 mg/kg of TMR) at start of trial, AI time and 50 days post-AI.

Parameter	Group	Start of Trial	AI Time	50 Days Post-AI
Glucose (mmol/L)	CPS-0	4.01 ± 1.11	4.60 ± 0.84^c^	4.90 ± 0.64^c^
	CPS-2	4.08 ± 1.07	4.24 ± 0.23 ^b^	4.54 ± 0.13 ^b^
	CPS-4	4.07 ± 0.84 ^B^	6.79 ± 0.85 ^Aa^	7.79 ± 0.65 ^Aa^
	CPS-6	4.84 ± 0.90 ^B^	5.64 ± 0.80 ^Aabc^	5.94 ± 0.60 ^Aabc^
LPL (U/L)	CPS-0	636.58 ± 75.54	706.25 ± 60.58	756.62 ± 50.38
	CPS-2	531.57 ± 67.83 ^B^	702.81 ± 33.72 ^A^	757.21 ± 39.46 ^A^
	CPS-4	538.79 ± 98.52 ^B^	728.60 ± 58.21 ^A^	779.50 ± 68.12 ^A^
	CPS-6	632.86 ± 64.92	678.05 ± 105.27	728.21 ± 87.18
AST (mU/mL)	CPS-0	10.58 ± 1.16	9.46 ± 1.27	10.55 ± 0.35
	CPS-2	10.49 ± 1.49	11.88 ± 1.98	12.91 ± 0.87
	CPS-4	8.86 ± 1.68 ^B^	11.34 ± 0.95 ^A^	13.23 ± 1.05 ^A^
	CPS-6	11.43 ± 1.80	10.79 ± 0.91	11.88 ± 0.72

Note: Different small letters (^a,b,c^) denote the difference (*p* < 0.05) among the groups in same column and large (^A,B^) superscripts show the difference (*p* < 0.05) along same row.

## Data Availability

All the data and materials will be available on reasonable request from the corresponding author.

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
