# Peer review of "Dietary Supplementation of Capsaicin Enhances Productive and Reproductive Efficiency of Chinese Crossbred Buffaloes in Low Breeding Season"

_animals, 2022, doi:10.3390/ani13010118_

Round 1

Reviewer 1 Report (Previous Reviewer 1)

Dear authors,
I evaluated the manuscript identified as animals-2076581 (Dietary Supplementation of Capsaicin Enhances Productive and Reproductive Efficiency of Chinese Crossbred Buffaloes in Low Breeding Season).
I ask you for a minor revision, mainly with respect to the following points:
- control of text editing, i.e. checking throughout the text if acronyms are always mentioned at the first mention (see blood parameters, for example) and if punctuation is adequate (for example, see line 119), etc.;
- conclusions, for which it would be useful, in my opinion, to make a brief recall of the research objectives and link the conclusions to these. In addition, it would be desirable that the authors also define the future studies that may result from the results obtained.
I wish you good work

Author Response

Cover letter and response to reviewers 

We, the authors of manuscript entitled: “Dietary Supplementation of Capsaicin Enhances Productive and Reproductive Efficiency of Chinese Crossbred Buffaloes in Low Breeding Season”, hereby declare the manuscript is solely submitted to “Animals” for publication.

The article is not submitted or published in parts or as whole to any other scientific journal or magazine. The manuscript has been carefully read and structured according to the instructions designed by the Journal.

The article was previously submitted to Journal “Animals”; however, resubmission was suggested after proper revision.

Instructions to the authors have been read and all the authors are aware about the submission policy of the journal.

There is no known conflict of interest regarding financial or any other sort.

I declare that permission of animal ethics has been taken and included as a statement in the text. The mentioned work is original and is contribution of the listed authors.

Dear Editor

We are grateful for your precious time for conducting the early and smooth review process for submitted manuscript. We are also thankful for reviewers for their valuable comments and suggestions. Currently, we also tried the best efforts to improve the text expression and quality of manuscript for better readability and understandings.

Response to reviewer 1

I evaluated the manuscript identified as animals-2076581 (Dietary Supplementation of Capsaicin Enhances Productive and Reproductive Efficiency of Chinese Crossbred Buffaloes in Low Breeding Season).

I ask you for a minor revision, mainly with respect to the following points:
- control of text editing, i.e. checking throughout the text if acronyms are always mentioned at the first mention (see blood parameters, for example) and if punctuation is adequate (for example, see line 119), etc.;

Author response: We are also thankful for reviewers for their valuable comments and suggestions.The article has been revised and improved the English script and thoroughly revised the acronyms or punctuations in text. It has also been tried to omit the irrelevant materials in previous submission.

- conclusions, for which it would be useful, in my opinion, to make a brief recall of the research objectives and link the conclusions to these. In addition, it would be desirable that the authors also define the future studies that may result from the results obtained.

Author response: The revised conclusion of MS is more elaborative and clearly described the research objectives. The conclusions were based on significant data of production and reproductive indices.  

I wish you good work

Author response: Thank you so much for sparing time to review the article.We hope that revised manuscript contains satisfactory responses for the editor and the reviewers.

Reviewer 2 Report (New Reviewer)

The manuscript covers a subject matter that is topical and relevant to the industry. It is scientifically sound. The analyses and presentation of the results can however be improved. The authors should consider regression analyses, repeated measures analyses and possibly time series analyses as ways to analyze their data. Some of the data/results can be better understood and appreciated if presented graphically as well 

Author Response

Cover letter and response to reviewers 

We, the authors of manuscript entitled: “Dietary Supplementation of Capsaicin Enhances Productive and Reproductive Efficiency of Chinese Crossbred Buffaloes in Low Breeding Season”, hereby declare the manuscript is solely submitted to “Animals” for publication.

The article is not submitted or published in parts or as whole to any other scientific journal or magazine. The manuscript has been carefully read and structured according to the instructions designed by the Journal.

The article was previously submitted to Journal “Animals”; however, resubmission was suggested after proper revision.

Instructions to the authors have been read and all the authors are aware about the submission policy of the journal.

There is no known conflict of interest regarding financial or any other sort.

I declare that permission of animal ethics has been taken and included as a statement in the text. The mentioned work is original and is contribution of the listed authors.

Dear Editor

We are grateful for your precious time for conducting the early and smooth review process for submitted manuscript. We are also thankful for reviewers for their valuable comments and suggestions. Currently, we also tried the best efforts to improve the text expression and quality of manuscript for better readability and understandings.

Response to reviewer 2

The manuscript covers a subject matter that is topical and relevant to the industry. It is scientifically sound.

The analyses and presentation of the results can however be improved.

Author response: Thanks for your suggestions.In view of reviewer comments, the data were reanalysed. Also modified the expression of results according to observed data. 

The authors should consider regression analyses, repeated measures analyses and possibly time series analyses as ways to analyze their data.

Author response: The whole data of manuscript has been re-analyzed in light of reviewers’ recommendations.

Some of the data/results can be better understood and appreciated if presented graphically as well 

Author response: The results section was presented in tables and figures in light of reviewer observations.

Round 2

Reviewer 2 Report (New Reviewer)

Satisfactory responses to issues raised

This manuscript is a resubmission of an earlier submission. The following is a list of the peer review reports and author responses from that submission.

Round 1

Reviewer 1 Report

General remarks

Dear authors,

I evaluated the manuscript identified as animals-1953566.

The study summarizes the effects of capsaicin dietary supplementation on the productive and reproductive performance of buffaloes under heat-stress conditions. In this context, the results proposed by the authors appear extremely interesting and certainly deserve of being disclosed to the scientific community. However, I have been able to detect several inaccuracies that deserve to be carefully reviewed by the authors. My major concerns regard the materials and methods and results. My suggestions are detailed below, section by section. Hoping to have contributed to improving the manuscript quality. Good works.

Specific comments

L 16-20: In my opinion, the authors should improve the simple summary. Indeed, it is vitally important that scientists can describe their work simply and concisely to the public, especially in an open-access online journal. According to the journal template, the simple summary should contain, rather than the aims, objectives, and pertinent results, a clear statement of the problem addressed and how they will be valuable to society. This should be written for a lay audience, i.e., no technical terms without explanations. In addition, no abbreviations should be used. Thanks.

L 22 (and through the manuscript): can the authors explain what they mean by "litters"? What relevance does the term have with milk production? Thanks.

L 26-29: to gain readability, the authors can re-phase the description of the feeding treatments avoiding redundancies. Thanks.

L 30 (and through the manuscript): please, the authors are requested to uniform the "P" to the standards of the journal by reporting it in lowercase and italics. Thanks.

L 44: authors are advised to carefully check the journal template for the abbreviation list. Normally, this journal requires specifying the abbreviations at the first mention, while no indication is made about the list. In my opinion, the list of abbreviations proposed by the authors is extremely useful. However, it may not be in line with the newspaper's standards. I invite you to check again thanks.

L 52-53: these claims are also widely supported by doi.org/10.3390/ani11092683. Authors are requested to update. Thanks.

L 53 (and through the manuscript): I point out to the authors that the journal on which they aspire to publish adopts a different style of references to the one used. Therefore, I suggest consulting the template and standardizing the bibliography according to the required standards. Thanks.

L 57: the reference “Giuseppe et al., 2005” is not among the listed references. Authors are advised to check. Thanks.

L 99-103: The ethical statement should be made explicit in a specific paragraph at the end of the text. Authors are requested to check the journal template. Thanks.

L 107-109: the buffalo's reproductive seasonality varies mainly in relation to the length of daily light, which in turn changes, on a global scale, on a seasonal level according to the different breeding areas. As reported by the authors, in the study area this period falls between July and August, or when, after the solstice, the hours of daylight begin to decrease (at least in the temperate climate areas). To avoid localisms but, at the same time, make the proposed results more useful for a large audience, I believe it may be appropriate to highlight the photoperiodic regime that occurred during the test. As written, it seems that the authors refer exclusively to temperature as the only environmental factor of low reproduction. Thanks.

L 132: with respect to what scale can it be said that the buffaloes had a BCS? In my opinion, I think it is useful for the authors to report the body condition scoring scale used. Thanks.

L 137: to the best of my knowledge, there are no NRC nutritional recommendations available for the buffalo species. I ask the authors, therefore, to clarify which nutritional standards they used to formulate the experimental diets. From another point of view, it might be more useful to consider other systems of expression of nutritional requirements. In this context, I suggest consulting doi.org/10.3390/ani10030515 as standard (which I recommend using as a reference). Thanks.

L 141: if possible, it would be advisable for the authors to indicate whether during the trial they sampled feeds and diets and, above all, according to which analytical methods they assessed the chemical composition depicted in table 1. Thanks.

L 142 (Table 1): in my opinion Table 1 has some inconsistencies. First, the header, since the table only shows the ingredients and chemical composition of the diet, while nothing is shown about the chemical composition of the concentrates. Then, no indication is given as to whether ingredients and chemical composition are referred to as feed basis or as dry matter basis. Finally, "distiller grains" should be indicated, consistently with the other ingredients, in capital letters. Thanks.

L 149: do the tested doses (mg/kg) of capsaicin refer to the kg of TMR dry matter or are reported as feed-basis?

L 184-189: the authors claim to have taken samples at the two daily milking to evaluate the milk's chemical composition (L 184-185). Subsequently (L 189) they state that they evaluate the milk sampled on the 20 and 30 trial days. The two statements seem incongruous in my opinion. Please clarify.

L 185: “litters”? Explain, please.

L 223 (Table 2): the levels of capsaicin supplementation depicted in the table (0, 2, 4, 6) are not consistent with those indicated in materials and methods (L 146-150, i.e., 0, 3, 6, 9). Conversely, the same levels indicated above are commented on in the description of the results (L 146-150). What is the rationale for these inconsistencies? The authors are requested to carefully check the adequacy of the information provided in all the tables and in the text including in the "results" and "discussion" sections. Thanks.

L 233 and L 250: see comments for L 223.

L 301: check uppercase, thanks.

L 353: are you sure that the average dose is 4-6 mg? See the comments made earlier, thanks.

L 377: the reference list does disagree with the journal's standards. Please check, thanks. Additionally, many references in the text are inconsistent with the list, as if the bibliography came from different manuscripts.

Author Response

Response to Reviewer 1 Comments

We appreciate your valuable suggestions. We have went through the revision and tried the best to address all comments in the current revision. The revised text has been highlighted in blue font. We hope that revised manuscript would have satisfied responses for the editor and the reviewers.

Reviewer 1

Open Review

General remarks

Dear authors,

I evaluated the manuscript identified as animals-1953566.

The study summarizes the effects of capsaicin dietary supplementation on the productive and reproductive performance of buffaloes under heat-stress conditions. In this context, the results proposed by the authors appear extremely interesting and certainly deserve of being disclosed to the scientific community. However, I have been able to detect several inaccuracies that deserve to be carefully reviewed by the authors. My major concerns regard the materials and methods and results. My suggestions are detailed below, section by section. Hoping to have contributed to improving the manuscript quality. Good works.

Response: Thanks for encouraging comments. Highlighted points were highly beneficial while revising the text of manuscript. The manuscript has been revised in light of reviewers’ observation and suggestions. The revised manuscript is clearer for scientific community.

Specific comments

L 16-20: In my opinion, the authors should improve the simple summary. Indeed, it is vitally important that scientists can describe their work simply and concisely to the public, especially in an open-access online journal. According to the journal template-..,m, the simple summary should contain, rather than the aims, objectives, and pertinent results, a clear statement of the problem addressed and how they will be valuable to society. This should be written for a lay audience, i.e., no technical terms without explanations. In addition, no abbreviations should be used. Thanks.

Response: The simple summery has been revised to make simple and easy for readers.

L 22 (and through the manuscript): can the authors explain what they mean by "litters"? What relevance does the term have with milk production? Thanks.

Response: It has been revised as liters (unit of volume).

L 26-29: to gain readability, the authors can re-phase the description of the feeding treatments avoiding redundancies. Thanks.

Response: Thanks for highlighting this point. It has been revised in M&M section.

L 30 (and through the manuscript): please, the authors are requested to uniform the "P" to the standards of the journal by reporting it in lowercase and italics. Thanks.

Response: Revised according to suggestion.

L 44: authors are advised to carefully check the journal template for the abbreviation list. Normally, this journal requires specifying the abbreviations at the first mention, while no indication is made about the list. In my opinion, the list of abbreviations proposed by the authors is extremely useful. However, it may not be in line with the newspaper's standards. I invite you to check again thanks.

Response: It has been deleted.

L 52-53: these claims are also widely supported by doi.org/10.3390/ani11092683. Authors are requested to update. Thanks.

Response: Recent citations were included.  

L 53 (and through the manuscript): I point out to the authors that the journal on which they aspire to publish adopts a different style of references to the one used. Therefore, I suggest consulting the template and standardizing the bibliography according to the required standards. Thanks.

Response: Followed the journal style to cite the literature.  

L 57: the reference “Giuseppe et al., 2005” is not among the listed references. Authors are advised to check. Thanks.

Response: Reference list has been revised.

L 99-103: The ethical statement should be made explicit in a specific paragraph at the end of the text. Authors are requested to check the journal template. Thanks.

Response: Ethical statement has been placed at the end of text. 

L 107-109: the buffalo's reproductive seasonality varies mainly in relation to the length of daily light, which in turn changes, on a global scale, on a seasonal level according to the different breeding areas. As reported by the authors, in the study area this period falls between July and August, or when, after the solstice, the hours of daylight begin to decrease (at least in the temperate climate areas). To avoid localisms but, at the same time, make the proposed results more useful for a large audience, I believe it may be appropriate to highlight the photoperiodic regime that occurred during the test. As written, it seems that the authors refer exclusively to temperature as the only environmental factor of low reproduction. Thanks.

Response: In M&M section, Subheading 2.1. Experimental Site and Climatic Conditions ….. has been revised in light of reviewers observations.

L 132: with respect to what scale can it be said that the buffaloes had a BCS? In my opinion, I think it is useful for the authors to report the body condition scoring scale used. Thanks.

Response: In M&M section, Subheading 2.2. Husbandry practices ……. Now clearly showed the BCS and other general observations about included buffaloes.

L 137: to the best of my knowledge, there are no NRC nutritional recommendations available for the buffalo species. I ask the authors, therefore, to clarify which nutritional standards they used to formulate the experimental diets. From another point of view, it might be more useful to consider other systems of expression of nutritional requirements. In this context, I suggest consulting doi.org/10.3390/ani10030515 as standard (which I recommend using as a reference). Thanks.

Response: We agreed the observation of reviewer. A recently recommended reference for buffalo TMR feeding has been added in text and reference list.

L 141: if possible, it would be advisable for the authors to indicate whether during the trial they sampled feeds and diets and, above all, according to which analytical methods they assessed the chemical composition depicted in table 1. Thanks.

Response: The chemical composition of ration was assessed through proximate analysis in feed samples prior and mid of feeding trial.

L 142 (Table 1): in my opinion Table 1 has some inconsistencies. First, the header, since the table only shows the ingredients and chemical composition of the diet, while nothing is shown about the chemical composition of the concentrates. Then, no indication is given as to whether ingredients and chemical composition are referred to as feed basis or as dry matter basis. Finally, "distiller grains" should be indicated, consistently with the other ingredients, in capital letters. Thanks.

Response: The Table 1 has been revised. Table 1 show the feed composition of ration on feed basis and proximate analysis was performed for TDN, CP, CF and EE. In addition, ingredients names were corrected as indicated.

L 149: do the tested doses (mg/kg) of capsaicin refer to the kg of TMR dry matter or are reported as feed-basis?

Response: Thanks for raising this important point. In the present study, CPS doses were also adjusted to kg of TMR as feed basis rather than dry matter base.

L 184-189: the authors claim to have taken samples at the two daily milking to evaluate the milk's chemical composition (L 184-185). Subsequently (L 189) they state that they evaluate the milk sampled on the 20 and 30 trial days. The two statements seem incongruous in my opinion. Please clarify.

Response: These lines were presented separately for milk composition analysis and milk production recordings. It is clearer for readers.

L 185: “litters”? Explain, please.

Response: It has been revised as liters (unit of volume).

L 223 (Table 2): the levels of capsaicin supplementation depicted in the table (0, 2, 4, 6) are not consistent with those indicated in materials and methods (L 146-150, i.e., 0, 3, 6, 9). Conversely, the same levels indicated above are commented on in the description of the results (L 146-150). What is the rationale for these inconsistencies? The authors are requested to carefully check the adequacy of the information provided in all the tables and in the text including in the "results" and "discussion" sections. Thanks.

Response: There were some typos while editing the text and dosages of CPS were presented correctly in revised text.

L 233 and L 250: see comments for L 223.

Response: Text has revised thoroughly.

L 301: check uppercase, thanks.

Response: Modified the lines.

L 353: are you sure that the average dose is 4-6 mg? See the comments made earlier, thanks.

Response: Thanks for comments. It has been modified thoroughly.

L 377: the reference list does disagree with the journal's standards. Please check, thanks. Additionally, many references in the text are inconsistent with the list, as if the bibliography came from different manuscripts.

Response: The reference list has been revised as per journal author’s instructions.

Reviewer 2 Report

1. In the introduction, please provide relevant literature to support the application background and research progress of capsaicin as a feed additive to feed cattle.

2. In this study, the experimental buffaloes were grouped according to different CPS doses, and the number was different. What is the basis for such a grouping method?

3. The CPS groups mentioned in Section 2.4 are 0 (CPS-0; n=26), 3 (CPS-3; n=22), 6 (CPS-6; n=25), or 9 (CPS-9; n=27) mg/kg of TMR daily which were inconsistent with the abstract?

4. Table 2 in Part 3.1 of this paper shows that the results are inconsistent with that in the abstract (CPS-6 results mentioned in the abstract are actually the results in the CPS-4 group). There are still such discrepancies in the following

5. Under the condition of heat stress, the intake of buffalo decreased, whether different doses of CPS had an impact on the DIM of buffalo, whether the parity of buffalo was similar, and whether different parity had an impact on the utilization of CPS.

6. The discussion part explained the experimental results, and whether the authors can further clarify the potential mechanism of capsaicin's significant impact on buffalo metabolism.

7. It is suggested that the key nutrients of capsaicin can be further discussed.

8. Capsaicin has an antibacterial effect, whether it has a certain influence on milk yield, milk composition, and reproduction during the experiment.

Author Response

Response to Reviewer 2 Comments

We appreciate your valuable suggestions. We have went through the revision and tried the best to address all comments in the current revision. The revised text has been highlighted in blue font. We hope that revised manuscript would have satisfied responses for the editor and the reviewers.

Reviewer 2.

  1. In the introduction, please provide relevant literature to support the application background and research progress of capsaicin as a feed additive to feed cattle.

Response: Last para of introduction section has been revised by adding few lines and references.

  1. In this study, the experimental buffaloes were grouped according to different CPS doses, and the number was different. What is the basis for such a grouping method?

Response: It has been revised that occurred due to some typos while editing the text. In Material and Methods section: sub-heading 2.3. CPS supplementation…. has been revised to remove any confusion.  

  1. The CPS groups mentioned in Section 2.4 are 0 (CPS-0; n=26), 3 (CPS-3; n=22), 6 (CPS-6; n=25), or 9 (CPS-9; n=27) mg/kg of TMR daily which were inconsistent with the abstract?

Response: The text has been revised and now showing the proper layout

  1. Table 2 in Part 3.1 of this paper shows that the results are inconsistent with that in the abstract (CPS-6 results mentioned in the abstract are actually the results in the CPS-4 group). There are still such discrepancies in the following

Response: It has been revised carefully.

  1. Under the condition of heat stress, the intake of buffalo decreased, whether different doses of CPS had an impact on the DIM of buffalo, whether the parity of buffalo was similar, and whether different parity had an impact on the utilization of CPS.

Response: In the present study, we did not record the feed intake level but it was ensured by that each buffalo should consume the provided CPS dose in TMR. Same parity and age buffaloes were included; therefore, it was expected that all buffaloes consumed the feed uniformly.

  1. The discussion part explained the experimental results, and whether the authors can further clarify the potential mechanism of capsaicin's significant impact on buffalo metabolism.

Response: Few lines have been included in discussion to describe the link of CPS consumption and buffalo metabolism.  

  1. It is suggested that the key nutrients of capsaicin can be further discussed.

Response: First para of discussion section is added to highlight the importance of CPS.  

  1. Capsaicin has an antibacterial effect, whether it has a certain influence on milk yield, milk composition, and reproduction during the experiment.

Response: We did not observed the antibacterial effect but low SCC in milk of CPS supplemented groups showed the potential effect of CPS as antibacterial agent. 

Round 2

Reviewer 1 Report

Dear authors,
I have evaluated the revised version of the manuscript identified as animals-1953566 and, in light of the changes made, I have no doubt about the validity of the manuscript, which I will recommend for publication. However, only a few clarifications are needed:
- in accordance with the journal standard, acronyms should be avoided in the simple summary (see L 20-21 of the new version);
- in my opinion, production should be expressed in liters per head (L 24);
- I would avoid starting a sentence with an acronym without preceding it with the article (L 71).

That's all. Congratulations and good luck.

Author Response

We appreciate for your valuable suggestions for this revision. We have addressed all the previous concerns raised by reviewers during previous versions. The revised text has been highlighted for proper review and tried to modify the text thoroughly for grammatical or expression mode. We did the best efforts to improve the quality of MS for better readability and understandings.  

Response to Reviewer 1 Comments

Dear authors,
I have evaluated the revised version of the manuscript identified as animals-1953566 and, in light of the changes made, I have no doubt about the validity of the manuscript, which I will recommend for publication. However, only a few clarifications are needed:
- in accordance with the journal standard, acronyms should be avoided in the simple summary (see L 20-21 of the new version);

Author response: It have been revised as highlighted by reviewers.

- in my opinion, production should be expressed in liters per head (L 24);

Author response: We revised as suggested

- I would avoid starting a sentence with an acronym without preceding it with the article (L 71).
Author response: Followed as indicated

That's all. Congratulations and good luck

Author response: Thank you so much for your kind cooperation.We hope that revised manuscript contain satisfactory responses for the editor and the reviewers.

Reviewer 2 Report

The revised manuscript has addressed all my concerns.

Author Response

We appreciate for your valuable suggestions for this revision. We have addressed all the previous concerns raised by reviewers during previous versions. The revised text has been highlighted for proper review and tried to modify the text thoroughly for grammatical or expression mode. We did the best efforts to improve the quality of MS for better readability and understandings.

Response to Reviewer 2 Comments

The revised manuscript has addressed all my concerns.

Author response: We are grateful for spending precious time for reviewing the manuscript, your suggestions made the revised text more attractive and simple for readers.  
